# Genome-Wide Identification and Functional Analysis of the *GASA* Gene Family Responding to Multiple Stressors in *Canavalia rosea*

**DOI:** 10.3390/genes13111988

**Published:** 2022-10-31

**Authors:** Mei Zhang, Zhengfeng Wang, Shuguang Jian

**Affiliations:** 1Guangdong Provincial Key Laboratory of Applied Botany, South China Botanical Garden, Chinese Academy of Sciences, Guangzhou 510650, China; 2Key Laboratory of South China Agricultural Plant Molecular Analysis and Genetic Improvement, South China Botanical Garden, Chinese Academy of Sciences, Guangzhou 510650, China; 3Key Laboratory of Vegetation Restoration and Management of Degraded Ecosystems, South China Botanical Garden, Chinese Academy of Sciences, Guangzhou 510650, China; 4Key Laboratory of Carbon Sequestration in Terrestrial Ecosystem, South China Botanical Garden, Chinese Academy of Sciences, Guangzhou 510650, China

**Keywords:** *Gibberellic Acid-Stimulated Arabidopsis (GASA)*, ecological adaptability, *Canavalia rosea*

## Abstract

In plants, the *Gibberellic Acid-Stimulated Arabidopsis* (*GASA*) gene family is unique and responds to ubiquitous stress and hormones, playing important regulatory roles in the growth and development of plants, as well as in the resistance mechanisms to biotic and abiotic stress. In this study, a total of 23 *CrGASA*s were characterized in *C. rosea* using a genome-wide approach, and their phylogenetic relationships, gene structures, conserved motifs, chromosomal locations, gene duplications, and promoter regions were systematically analyzed. Expression profile analysis derived from transcriptome data showed that *CrGASA*s are expressed at higher levels in the flowers or fruit than in the leaves, vines, and roots. The expression of *Cr**GASA*s also showed habitat- and environmental-stress-regulated patterns in *C. rosea* analyzed by transcriptome and quantitative reverse transcription PCR (qRT-PCR). The heterologous induced expression of some *Cr**GASA*s in yeast enhanced the tolerance to H_2_O_2_, and some *CrGASA*s showed elevated heat tolerance and heavy metal (HM) Cd/Cu tolerance. These findings will provide an important foundation to elucidate the biological functions of *CrGASA* genes, especially their role in the ecological adaptation of specific plant species to tropical islands and reefs in *C. rosea*.

## 1. Introduction

Plant-specific *Gibberellic Acid-Stimulated Arabidopsis* (*GASA*) genes, which exist extensively, encode a series of small cysteine-rich proteins with high evolutionary conservation that are characterized by a signaling amino acid region at their N-terminus and a conserved cysteine-rich domain (GASA domain) at the C-terminus [1,2]. GASA proteins are also named snakin, because the early-discovered Snakin-1 (SN1) from potato (*Solanum tuberosum*) shares two conserved regions with snake venom kistrin, even though it typically lacks the residues responsible for the desintegrin action of the venom [3,4]. Previously, the *GASA* gene was primarily called *GAST* (*Gibberellic Acid-Stimulated Transcript*) due to its expression being obviously induced by exogenous gibberellic acid (GA) in tomato (*GAST1*, *Solanum lycopersicum*) [5]. In *Petunia hybrida*, this gene family was also called the *Gibberellin-Induced gene from Petunia (GIP)* [6], and in rice it was named *Gibberellic Acid-Stimulated Rice (OsGASR)* [7]. A typical feature of GASA proteins is that they all have 12 conserved cysteine residues at constant positions at the C-terminal region, while a putative signal peptide at the N-terminus is closely followed by a variable hydrophilic region in the middle part. The GASA domain consists of a highly conserved 12-Cys motif, “XnCX_3_CX_2_RCX_8 (9_)CX_3_CX_2_CCX_2_CXCVPXGX_2_GNX_3_CPCYX_10(14)_KCP,” in which R is arginine, V is valine, P is proline, G is glycine, Y is tyrosine, K is lysine, and X represents any of the other 20 amino acids, except for cysteine [1].

Plant *GASA*s have been proven to play multiple potential roles in plant growth and development, particularly flower induction and seed development [6,8,9]. GASA proteins participate in hormonal signaling pathways, mainly gibberellic acid, and including other hormones, such as ethylene, brassinosteroid, abscisic acid, jasmonic acid, and salicylic acid [10,11,12,13,14,15,16]. GASA proteins are a type of plant antimicrobial peptide (AMP) and play different roles in response to a variety of pathogens, including bacteria, fungi, and even nematodes [4,17,18,19]. Moreover, evidence has demonstrated that GASA proteins could participate in abiotic stress tolerance, probably by acting as cellular integrators or playing roles in redox regulation [12,20,21,22].

Although many previous studies have shown that most *GASA* homologues are involved in numerous biological processes, mainly including the development of plant flowering, seed development, fruit growth, and cell elongation in vegetative organs [8,9,12,23,24,25], the study of plant *GASA* genes is of great significance for elucidating the molecular mechanisms of defense and stress resistance in plants [12]. Due to sessile features, plants must encounter various abiotic stressors, including salt, drought, heavy metals, extreme temperatures, chemical toxicity, and nutrient deficiencies [26]. To survive these abiotic stressors, plants have developed a series of physiological, molecular, and biochemical strategies, including enhancing the content of antioxidants, accumulating cellular stress proteins as molecular chaperones, inducing the expression of resistance genes, or adjusting cellular constituents or compatible solutes. Many previous studies have shown that GASAs are a group of environmental-stress-responsive proteins that play crucial roles both in protecting against insects and disease and in regulating multiple abiotic stressors [27].

The first plant *GASA* gene was identified from tomato in 1992 [5], and the first functionally defined plant GASA/snakin peptide was potato Snakin-1 (StSN1) from its tubers [3]. An increasing number of plant *GASA* genes have been cloned and systematically studied in a variety of plant species, forming pathogen resistance and plant developmental process points [19,28]. The Arabidopsis *GASA* gene family consists of 15 members, from *GASA1* to *GASA15* [2,28], and only five members (*GASA4*, *5*, *6*, *10*, and *14*) have been characterized, with functions including phytohormonal signaling integrators, interaction partners of protein complexes, and metalloproteins with antioxidant capacity [29,30]. In studies of abiotic stress responses, different *GASA* genes have presented specific roles. For example, overexpression of *AtGASA4* in Arabidopsis could enhance tolerance to heat stress, probably by suppressing the accumulation of reactive oxygen species (ROS) [21]. The expression of *AtGASA4* and *AtGASA6* was generally upregulated by growth hormones (auxin, BR, cytokinin, and GA) and downregulated by stress hormones (ABA, JA, and SA), while the overexpression of *AtGASA6* caused early flowering in Arabidopsis [16]. Conversely, overexpression of *AtGASA5* in Arabidopsis can increase its sensitivity to heat stress [22], and this gene may be suppressed by GA [15]. AtGASA14 is a specific member due to its extraordinary proline-rich protein (PRP) domain in the N-terminal region. Overexpression of *AtGASA14* in Arabidopsis showed elevated abscisic acid (ABA) and salt tolerance and a better ability to scavenge ROS than wild-type plants [12]. The *Gerbera hybrida GASA* genes *GIP2*, *GIP4,* and *GIP5* are induced by H_2_O_2_, and the overexpression of *GIP2* in transgenic petunia shows reduced H_2_O_2_ levels after osmotic stress or ABA treatment [20]. Rice *GASA* genes’ expression levels were regulated by different abiotic stressors, including salt, drought, cold, heat, and metal stress [28]. In *Eucalyptus globulus*, the *Snakin-2* gene was upregulated by metal stress combined with infection by the rhizospheric microorganism *Chaetomium cupreum* [31]. In general, systematic research on *GASA* genes’ involvement in abiotic stress responses is limited, but research has indicated that the GASA gene family might be significantly involved in responding to external stimuli and environmental suitability in plants.

Until now, many *GASA* gene families have been identified using whole-genome analyses in a wide variety of plant species [2]. Maize possesses 10 GASA members [32], and at least 10 OsGASR members were identified in the rice *GASA* family [28]. The *GmGASA* family has 37 members [33] in the ancient tetraploid soybean, while, in the common wheat genome (*Triticum aestivum*, hexaploid), there are 37 designated *GASA* genes [34]. Given the importance of plant GASAs as key growth regulators and antimicrobial peptides in some non-food cash crops, *GASA*s have attracted increasing attention in studies of plant–pathogen resistance, stress tolerance, and growth regulation. For example, in the apple (*Malus domestica*) genome, there are 26 *MdGASA* genes, and some members are involved in flower induction [35]. Poplar is an important tree species for shelterbelt and timber forests, and there are 21 candidate *GASA*s in *Populus trichocarpa* and 19 in *Populus euphratica*. Some members are widely involved in hormone responses, growth, and development for vegetative organs, and drought stress responses [36,37]. Grapevine (*Vitis vinifera*) is a major fruit crop, and the demand for seedless grapes is greatly increasing. Some *VvGASA* genes might play roles during different phases of seed development and in different tissues in seedless grape cultivars [38]. The *Citrus clementina* canker pathogen *Xanthomonas citri* is the primary cause of citrus loss; some *GASA*s in the 18-member *CcGASA* family are speculated to respond to *X. citri* infection [39]. Tobacco (*Nicotiana tabacum*) has 18 *GASA* genes, and some *NtGASA*s display unique or distinct expression patterns in different tissues, suggesting their potential roles in tobacco plant growth and development [40].

*Canavalia rosea* is a perennial twining herb distributed in the semi-arid and saline–alkali areas of coastal regions; it has shown great halotolerance and multiple resistance to many environmental adversities, including drought, salinity/alkaline, heat, and low nitrogen and phosphorous stress. Additionally, as a mangrove-associated species distributed in tropical and subtropical areas, *C. rosea* also has the potential to absorb or enrich heavy metals (HMs) at beaches or near river estuaries. Therefore, *C. rosea* can be used as a pioneer species with good wind-breaking and sand-fixing in the ecological reconstruction of tropical coral reefs or for phytoremediation with the protection of coastal belts. It is of particular interest to identify the gene-regulatory network involved in environmental adaptation in *C. rosea*. The *GASA* gene family has been confirmed to be involved in numerous physiological and biological processes, displaying complex and diverse functions.

In this study, we performed the genome-wide characterization of the *GASA* genes in *C. rosea* to explore the potential roles of these family members in the adaptation of *C. rosea* to tropical coastal regions or coral islands. In total, 23 putative *CrGASA* genes were identified in the *C. rosea* genome and subjected to phylogenetic, gene structure, motif, and chromosomal location analyses. The tissue-specific expression profiles, as well as the differential expression profiles of *CrGASA*s under abiotic stress, were also analyzed. Moreover, several *CrGASA* genes were cloned and expressed in yeast for the further identification of the *CrGASA*s’ functions in stress tolerance. Our results provide a platform for the further investigation of the functions of *CrGASA*s in the adaptation of *C. rosea* to multiple stress conditions on tropical coral islands and reefs, as well as providing significant insights into the function of plant *GASA*s as promising candidate genes for breeding HM-related phytoremediation.

## 2. Materials and Methods

### 2.1. Plant Materials and Stress Treatments

The seeds of *C. rosea* were collected from the coastal regions of Hainan Province, China, in 2019. *C. rosea* seedlings were cultivated under normal conditions (22 °C, with a photoperiod of 16 h light/8 h darkness) in the South China Botanical Garden (SCBG, 23°18′76″ N, 113°37′02″ E). The mature leaf samples were taken from perennial *C. rosea* plants growing on Yongxing Island (YX, 16°83′93′′ N, 112°34′00″ E) and SCBG. One-month-old *C. rosea* seedlings were used for various abiotic or heavy metal (HM) stress treatments, and the leaves and roots were collected separately after stress treatment, immediately frozen in liquid nitrogen, and stored at −80 °C for subsequent use. In brief, the seedlings were soaked in 600 mM NaCl, 150 mM NaHCO_3_ (pH 8.2), 300 mM mannitol, and 45 °C pre-warmed 1/2 Hoagland solution for high salinity, alkaline, high osmotic, and heat stress, respectively. For HM stress, *C. rosea* seedlings were subjected to 0.1 mM CdCl_2_, 0.5 mM ZnSO_4_, 1 mM MnCl_2_, and 0.1 mM CuSO_4_ solutions with the roots submerged. Plant tissues were collected at different time points (2 and 48 h for RNA-seq and qRT-PCR). Three independent biological replicates were used.

### 2.2. Identification, Ka/Ks Calculation, and Evolutionary Analyses of the CrGASA Family in C. rosea

Whole-genome sequencing was performed, and the whole genomic DNA sequence information was submitted to GenBank (Accession No.: JACXSB000000000), which will be released on 16 September 2024. The assembled genome data of *C. rosea* were annotated with the InterPro [41] and Pfam [42] databases for gene identification and DIAMOND [43] and InterProscan [41] for all *C. rosea* protein information with conserved domains and motifs (e < 1 × 10^−5^). The Pfam ID (GASA domain, PF02704) was used to search for CrGASA family members, and putative sequences of CrGASA proteins were identified and submitted to the NCBI Conserved Domain Database (https://www.ncbi.nlm.nih.gov/Structure/cdd/wrpsb.cgi (accessed on 1 March 2022)) to confirm the presence of the GASA domain.

The nucleotide and protein sequences of CrGASA members were tracked from the genome database of *C. rosea*, and the gene names were appointed according to their positions on the 11 chromosomes (from chromosomes 1 to 11 and one scaffold, CrGASA1 to CrGASA23). Then, the isoelectric point, number of amino acids, and molecular weight of CrGASAs were predicted using the ExPASy tool (http://web.expasy.org/protparam/ (accessed on 1 March 2022)). The protein sequences of GASAs from Arabidopsis (15 AtGASAs) [28], rice (11 OsGASRs; note that, compared with the related reference, we added a new putative rice GASA protein by Pfam search, LOC_Os03g14550.1, as OsGASR1, and the other ten OsGASRs listed in the reference were correspondingly designated as OsGASR2-11) [28], and soybean (37 GmGASAs) [33] were used to construct a phylogenetic tree using MEGA 6.0 with the neighbor-joining (NJ) method and a bootstrap test of 1000 replicates. The Gene Structure Display Server (http://gsds.cbi.pku.edu.cn (accessed on 1 March 2022)) was used to predict the exon/intron structures of each CrGASA. The conserved motifs of CrGASA proteins were analyzed using the Multiple Em for Motif Elicitation (MEME) software (http://meme-suite.org/tools/meme (accessed on 1 March 2022)), with a maximum of 10 motifs and a maximum width of between 6 and 50 amino acids.

### 2.3. Chromosomal Location and Sequence Alignments

All *CrGASA*s were mapped to 11 *C. rosea* chromosomes and scaffolds based on physical location information from the database for *C. rosea* genomic sequence information using MG2C 2.1 (http://mg2c.iask.in/mg2c_v2.1/ (accessed on 1 March 2022)). The synonymous and non-synonymous substitution rates (Ks and Ka, respectively) and the probability (*p*-value) of Fisher’s exact test of neutrality were calculated to explore the selective pressures on the duplication of *CrGASA*s based on all nucleotide sequences, using the Nei–Gojobori model with 1000 bootstrap replicates [44]. Gene segmental duplication events of the *CrGASA* family were analyzed using the MCScanX software (http://chibba.pgml.uga.edu/mcscan2/ (accessed on 1 March 2022)), and tandem duplications were identified manually. The protein sequences of the CrGASAs were aligned using ClustalW (http://www.clustal.org/ (accessed on 1 March 2022)) and visualized with GeneDoc (https://github.com/karlnicholas/GeneDoc (accessed on 1 March 2022)). The WebLogo platform (http://weblogo.berkeley.edu/logo.cgi (accessed on 1 March 2022)) was used to generate and analyze the sequence logos.

### 2.4. Cis-Regulatory Element Analysis of CrGASA Promoters

The predicted *cis*-regulatory elements were scanned using the PlantCARE program (http://bioinformatics.psb.ugent.be/webtools/plantcare/html/ (accessed on 1 March 2022)), searching the promoter regions (1000 bp upstream from the translation start site) of all *CrGASA*s. After sorting the *cis*-regulatory elements obtained from PlantCARE, the results were visualized and mapped to the *CrGASA* promoters using TBtools software [45].

### 2.5. RNA-Seq of Different C. rosea Tissues under Different Stress Treatments

Tissue-specific expression profile analysis for *CrGASA*s during different developmental stages of *C. rosea* was conducted using Illumina HiSeq X sequencing technology. Five different tissues from *C. rosea* plants (root, vine, young leaf, flower bud, and young silique samples) were collected from *C. rosea* adult plants and seedlings growing in the SCBG; mature leaf samples from *C. rosea* growing in SCBG and on YX Island were examined using FastQC (http://www.bioinformatics.babraham.ac.uk/projects/fastqc/ (accessed on 1 March 2022)) based on the primary 40 Gb clean reads and were mapped to the *C. rosea* reference genome using Tophat v.2.0.10 (http://tophat.cbcb.umd.edu/ (accessed on 1 March 2022)). For the expression profiles of *CrGASA*s under different abiotic stress treatments, *C. rosea* seedling tissues (including leaves and roots) were also collected and sequenced at the transcriptome level. Then, all EST information was mapped to the *C. rosea* reference genome. Gene expression levels were calculated as fragments per kilobase (kb) of transcript per million mapped reads (FPKM) according to the length of the gene and the read counts mapped to the gene: FPKM = total exon fragments/[mapped reads (millions) × exon length (kb)]. The expression levels of *CrGASA*s were visualized as clustered heatmaps (log2) using TBtools, which were directly shown with FPKM values by Microsoft Excel 2010.

### 2.6. Expression Pattern Analysis Using Quantitative Reverse Transcription (qRT)-PCR

The total RNA was extracted from the same tissues for RNA-seq assays using a plant RNA extraction kit (TIANGEN, Beijing, China), and cDNA was synthesized using AMV reverse transcriptase (TransGen Biotech, Beijing, China) according to the manufacturer’s instructions. The total RNA and cDNA concentration and quality were tested using a NanoDrop 1000 (Thermo Fisher Scientific, Waltham, MA, USA), with the integrity of the total RNA being checked on 1% agarose gel. qRT-PCR was used to analyze the relative expression levels of the *CrGASA*s. The housekeeping gene *CrEF-1α* was used as a reference. The sequences of the primers are listed in Appendix A. In brief, the qRT-PCR was carried out for six selected genes (*CrGASA**2*, *CrGASA**3*, *CrGASA**7*, *CrGASA**13*, *CrGASA**14*, and *CrGASA**16*) using 2 × Ultra SYBR Green qPCR Mix (CISTRO BIO, Guangzhou, China) on a LightCycler480 system (Roche, Basel, Switzerland), according to the manufacturer’s instructions. The relative expression levels of the members of the CrGASA were displayed with the 2^−∆∆^^CT^ method.

### 2.7. Functional Identification with a Yeast Expression System

The open reading frames (ORFs) of the *CrGASA*s were PCR-amplified from different cDNA samples of *C. rosea* with gene-specific primer pairs (listed in Appendix A). After several PCR procedures, the PCR fragments were purified and cloned into the *Bam*HI and *Eco*RI sites of pYES2 to yield recombinant plasmids of CrGASAs-pYES2 and sequenced. The *Saccharomyces cerevisiae* wild-type (WT) strain BY4741 (Y00000) and six deletion mutant strains *skn7**Δ* (Y02900), *ycf1**Δ* (Y04069), *pmr1**Δ* (Y04534), *cot1**Δ* (Y01613), *smf1**Δ* (Y06272), and *cup2**Δ* (Y04533) were obtained from Euroscarf (http://www.euroscarf.de (accessed on 1 March 2022)). The double-mutant strain *zrc1Δ*/*cot1Δ* was obtained from Sanders’ lab [46]. The plasmids were introduced into yeast using the LiAc/PEG method. These CrGASA proteins will be expressed under the induction of 2% galactose in medium, with the control of galactose-induced promoter Pro_GAL1_. Yeast growth and metal sensitivity tests were performed as previously described [47]. Single colonies of yeast transformants were selected and used to inoculate a liquid synthetic drop-out uracil medium with 2% galactose (SDG-Ura) medium. It was then incubated overnight or longer at 30 °C, diluted with fresh pre-warmed SDG medium (volume ratio 1:10), and incubated with vigorous shaking for approximately 30 h at 30 °C to reach an optical density of 1 at OD600 (optical density at 600 nm). The cells were then serially diluted in 10-fold steps, and 2 μL aliquots of each were finally spotted onto SDG medium plates with or without HM stressors. To determine heat tolerance, the liquid yeast cultures were incubated in a constant-temperature bath (52 °C) at different durations (WT for 30 min and *skn7**Δ* for 15 min); then, the cultures were spotted on solid SDG medium plates. Plates were incubated at 30 °C for 2 to 5 days and photographed.

### 2.8. Statistical Analyses

All analyses were conducted at least in triplicate, with the results shown as the mean ± SD (*n* ≥ 3). The Excel 2010 (Microsoft Corporation, Albuquerque, NM, USA) statistics program was used to perform statistical analyses.

## 3. Results

### 3.1. Overview of the C. rosea CrGASA Gene

In total, 23 *CrGASA* genes were identified from the *C. rosea* genome based on Pfam and NCBI Conserved Domain Database confirmation with other GASA proteins. *CrGASA* family sequence information is listed in Appendix A. In addition, based on chromosome localization, the 23 *CrGASA* genes were named *CrGASA1*–*CrGASA23*. Among them, *CrGASA1*–*CrGASA22* were located on nine chromosomes across the *C. rosea* genome (including 11 chromosomes), and only *CrGASA23* was located on one of the scaffolds, due to less-than-perfect genome assembly (Figure 1). Chromosomes 02 and 04 each contained four *CrGASA* genes, and chromosomes 06 and 07 each contained three genes. Chromosomes 03, 05, and 08 each had two *CrGASA* genes, while chromosomes 01 and 11 and an unassembled scaffold each contained only one. The physicochemical properties were estimated using the ExPASy server, and the CrGASAs’ length ranged from 70 (CrGASA19) to 233 aa (CrGASA17), and the MWs ranged from 7.86 to 24.55 kDa. The theoretical isoelectric point (pI) values were all greater than 8, and over half of the CrGASA members (13) presented a higher instability index (II) >40, indicating that these proteins are unstable in vivo. The aliphatic index ranged from 25.14 (CrGASA19) to 90.69 (CrGASA15), and most of the grand average hydropathicity values (GRAVY) were below 0, which indicated that although CrGASAs had different aliphatic amino acid content, most of the CrGASAs were hydrophilic. The transmembrane and 3D structure prediction showed that most of the CrGASAs had one or two transmembrane helices (TMHs), indicating their possible movement or redistribution across membranes. Moreover, the subcellular localization prediction that more than half of the CrGASA members were secretory proteins further supported the notion that CrGASAs are cellular shuttle proteins. The physicochemical characteristics of the CrGASAs are summarized in Table 1.

### 3.2. Phylogenetic Analysis and Multiple Sequence Alignment of CrGASA Members

To characterize the phylogenetic relationships among GASAs from different species, an unrooted NJ phylogenetic tree was constructed with 15 AtGASAs from Arabidopsis, 11 OsGASRs from rice, 37 GmGASAs from soybean, and 23 CrGASAs from *C. rosea*. All GASA proteins were divided into three groups (G1, G2, and G3) (Figure 2), which showed that there was a close relationship between the candidate CrGASA proteins in each of the three subgroups. In general, due to *C. rosea* being a leguminous plant species, the CrGASAs showed more close evolutionary relationships with GmGASAs from soybean (Leguminosae) than GASAs from Arabidopsis (Brassicaceae) and rice (Gramineae). There were more members of the G2 group in *C. rosea* than in most other plants [35,38,39], and the CrGASAs were almost evenly distributed in the three subgroups. Different from this, in the soybean genome, the G2 group number of GmGASA was also obviously smaller than that of G1 and G3 [33].

As shown in previous studies, most plant GASAs have a highly conserved C-terminal domain containing 12 conserved cysteines, designated as the GASA domain (containing 60 amino acids). Here, we aligned 23 CrGASAs’ conserved domains, in which we found that there were several variations in the positions of cysteines, including the second cysteine in CrGASA5, the eighth cysteine in CrGASA12, and the fifth cysteine in CrGASA20. Except for CrGASA12, the others ended with “KCP” at the C-terminus (Figure 3). After knocking out the genome sequencing factors, the GASA domains in the CrGASA family seemed to be more variable than in other plants [40].

The selection process history of the *CrGASA* genes has also been predicted through the Ka (non-synonymous substitution rate)/Ks (synonymous substitution rate) ratio, and the positive, negative (purifying), or neutral selection for *CrGASA* genes was considered when the ratio was >1, <1, or = 1, respectively. All segmental duplicates had Ka/Ks values less than 1 (Table 2), indicating that these gene pairs evolved under the influence of purifying selection. The distribution of segmental duplication of *CrGASA*s in *C. rosea* chromosomes is shown in Appendix A.

### 3.3. Analysis of CrGASA Proteins’ Conserved Motifs and Gene Structures

Four conserved protein motifs were identified in the CrGASA proteins, mainly due to their relatively short peptide chain lengths (70–233 aa). The highly conserved motifs 1 (KCARRCSKASRKKRCMRFCGTCCSKCKCVPPGTYGNKEEC) and 2 (CYNBLKTKGGKPKCP) merged to form the GASA domain, while motifs 3 (AKFLLVLILALIAISMLKTRVMASSADGC) and 4 (YGPGSLKSYQC) were more variable in different CrGASA members (Figure 4A,B).

Exon–intron structures were generated based on annotated *C. rosea* genome sequencing information using the Gene Structure Display Server program (Figure 4C). Even encoding relatively small proteins (70–233 aa), all *CrGASA* genes contain introns and some of them seem to be large (*CrGASA**22* and *CrGASA**23*), which is similar to that in other plant species [33,40]. Interestingly, in the highly homologous gene pairs *CrGASA18* and *CrGASA23*, *CrGASA3* and *CrGASA11*, and *CrGASA2* and *CrGASA16*, the gene structures were not very similar, and they had different numbers of exons.

### 3.4. Abiotic Stress-Related Cis-Regulatory Elements in CrGASA Promoters

To gain further insights into the regulatory mechanisms of *CrGASA* genes, a 1-kb promoter region upstream of the start codon (ATG) was isolated based on the *C. rosea* genome sequence and analyzed to identify potential *cis*-regulatory elements. These elements are listed in Figure 5 and Appendix A. Several plant hormone-related *cis* elements, including MeJA, auxin, gibberellin, salicylic acid, ethylene, and ABA, accounted for a larger portion of the total elements. Wound-responsive (WUN motif), defense and stress responsiveness (TC-rich repeats), and pathogen-inducible (as-1) elements were also widely distributed in these promoters, which might be closely related to *CrGASA*s’ function as defense-related genes. Moreover, *cis* elements involved in endosperm development (GCN4_motif) and meristem-specific expression (CAT-box) were identified in the promoters of several *CrGASA*s. Additionally, MYC- and MYB-binding sites were abundant, indicating that some *CrGASA*s were involved in drought-inducibility gene expression or other abiotic stressors (Figure 5). The presence of multiple *cis*-regulatory elements in *CrGASA*s’ promoter regions indicates the possible roles of this gene family in various physiological and biological processes, although this prediction needs to be further verified by experiments.

### 3.5. Tissue- and Habitat-Specific Expression Profiles of CrGASAs

The expression patterns of the 23 *CrGASA*s were investigated in different tissues or organs using RNA-seq to further verify their roles in regulating the growth and development of *C. rosea*. Five tissues or organs (root, vine, leaf, flower bud, and young fruit) from *C. rosea* plants were analyzed (Figure 6A). In general, the expression levels of most *CrGASA*s were relatively low in the roots, while, in the reproductive organs (flowers and fruit), *CrGASA*s presented higher expression levels overall. In the actively growing vines and leaves, some of the *CrGASA*s also showed high expression while being slightly lower than in flowers and fruit. The unique expression patterns showed that *CrGASA*s had strong spatiotemporal and tissue specificity in *C. rosea* plants, which might further suggest their obvious biological roles, most related to the vegetative growth and reproduction of *C. rosea*.

*Canavalia rosea* plants have a great advantage in adapting to multiple adversities on tropical coral islands or reefs, and plant *GASA* gene families have been found to be modulated by various abiotic stressors. In this regard, we performed a gene expression analysis of two mature leaf samples gathered from SCBG and YX Island (Figure 6B). The FPKM values for the above RNA-seq assays are listed in Appendix A. We noted that more than half of the *CrGASA*s had relatively low expression in both samples, while several members, such as *CrGASA5*, *CrGASA8*, *CrGASA9*, *CrGASA10*, *CrGASA11*, *CrGASA14*, *CrGASA17, CrGASA20*, and *CrGASA21*, showed relatively higher expression in the YX sample than in the SCBG sample, which indicated that these genes might play positive regulatory roles in this halophyte’s adaptation to coral island or reef habitats. Additionally, *CrGASA2* and *CrGASA16* showed higher expression in SCBG than in YX, suggesting that these two genes act on growth regulation, since the *C. rosea* plants showed better growth potential in SCBG under better care and with better nutrients than those on YX Island.

### 3.6. Expression Profile of CrGASA Genes in Response to Abiotic Stresses

The promoter analysis of *CrGASA*s suggested that these genes are involved in the response of *C. rosea* to different stressors. In this study, the transcriptional expression of *CrGASA*s in *C. rosea* seedlings was analyzed by RNA-seq after salt, alkaline, high osmotic, and heat stresses to mimic the extreme adversities of *C. rosea*’s original habitats on tropical coral islands or reefs. The expression patterns of *CrGASA*s under different challenges showed diverse changes compared to the controls. Overall, the expression levels of most *CrGASA*s in seedling leaves were much higher than those in roots, which might further indicate that CrGASAs are cellularly widely distributed, small proteins that work as protective molecules in ground substances and mainly play important roles in tissues’ exuberant growth. In general, after high salinity treatment, some *CrGASA*s were induced immediately in roots after 2 h, and in the challenge lasting 48 h, more than half of the *CrGASA*s were induced in roots and leaves. Accordingly, alkaline and high osmotic stress caused only a small portion of *CrGASA*s to be upregulated after 2 h and 48 h of treatment (Figure 7). Notably, 2 h heat stress also had an enormous influence on the expression levels of the *CrGASA*s, manifested in the mass inhibition of *CrGASA*s’ expression in roots, while obvious induction of most *CrGASA*s was expressed in the leaves. These data suggest that different *CrGASA*s may have specific functions, whereas the regulation patterns for *CrGASA*s could be tissue-specific and show multiple instances of functional differentiation.

As some plant *GASA* genes have been proven to be regulated by heavy metals [31,48], we further investigated the expression patterns of *CrGASA*s in response to different metal stressors, including cadmium (Cd), zinc (Zn), copper (Cu), and manganese (Mn). After 2 h of Cd treatment, the expression of *CrGASA3*, *CrGASA5*, *CrGASA10*, and *CrGASA15* was obviously induced in both root and leaf samples, while, after 48 h, the expression of the *CrGASA* family seemed to be inhibited, probably caused by *C. rosea* plant growth inhibition due to Cd toxicity. With the Zn and Cu stress challenges, the long-term treatments (48 h) resulted in the apparent upregulation of several *CrGASA*s, while the effects of 2 h of stress were indistinctive. Mn stress also induced the expression of some *CrGASA*s in the leaves (2 h) and roots (48 h) of *C. rosea* (Figure 8). The FPKM values for the above RNA-seq assays are listed in Appendix A.

### 3.7. qRT-PCR Analysis of CrGASAs

The expression levels of several *CrGASA*s were measured by qRT-PCR experiments to further determine whether transcript changes were associated with abiotic stress, mainly based on the RNA-seq results, and to choose the target *CrGASA*s with significant FPKM changes. Six *CrGASA*s were selected for qRT-PCR verification assays, and these assays were all repeated for three biological replicates. Here, the figures show only one typical experimental result of these biological replicates, which were all roughly consistent with the RNA-seq data (Figure 7, Figure 8, Figure 9 and Figure 10). Overall, alkaline stress caused the most apparent induction of *CrGASA2*, *CrGASA3*, *CrGASA13*, and *CrGASA14* in the *C. rosea* roots and *CrGASA2*, *CrGASA7*, *CrGASA14*, and *CrGASA16* in the leaves. Following high osmotic treatment (mannitol), all six *CrGASA*s showed induced expression either in roots or leaves at different time points (2 h or 48 h), although the long-term high osmotic challenge (48 h) decreased the expression of *CrGASA7* and *CrGASA13*. High salinity only temporarily induced the expression of *CrGASA2*, *CrGASA14*, and *CrGASA16*. Interestingly, the expression of most *CrGASA*s was not obviously affected by heat stress (Figure 9), which is similar to the RNA-seq results (Figure 7). Only *CrGASA7* and *CrGASA16* were significantly induced by the heat challenge.

### 3.8. Functional Characterization of CrGASAs in Yeast

To sufficiently investigate the function of *CrGASA*s in abiotic stress responses, the primary functions of *CrGASA*s was analyzed with a yeast heterologous expression system. The CrGASA proteins were induced in yeast cells in the galactose-supplied SDG medium. Given that plant GASAs have been shown to be involved in cellular redox homeostasis [27], although some reports have debated whether they are general or direct ROS scavengers [15], we checked their redox activities in yeast by monitoring their tolerance to hydrogen peroxide (H_2_O_2_). Although we initially chose the same six *CrGASA*s for further functional identification, the full-length cDNA of *CrGASA**3* was not successfully cloned and the subsequent work on *CrGASA**3* was ignored. The yeast H_2_O_2_-sensitive mutant stain *skn7*∆ and the wild type (WT) expressing five *CrGASA*s or only pYES2 (empty vector as a control) were grown in the presence of different H_2_O_2_ concentrations. *Skn7*∆ that expressed *CrGASA*s was significantly less sensitive to H_2_O_2_ than that only transformed with pYES2, although these *CrGASA*s were not completely complementary to the H_2_O_2_ sensitivity of *skn7*∆ compared with the WT (Figure 11A). Given that oxidative stress tolerance is considered as a universal trait for many abiotic or biotic stressors, it is reasonable that *CrGASA*s have been involved in environmental stress, such as salt/alkaline, dehydration, and heat stress, by exhibiting their antioxidant activity.

We next examined the HM tolerance of these five *CrGASA*s in yeast. As shown in Figure 11, the overexpression of *CrGASA14* both in the WT and Cd-sensitive mutant stain *ycf1*∆ could greatly elevate the Cd tolerance of these two yeast strains, while the other four *CrGASA*s could not (Figure 11B,C). We also detected other metal tolerances mediated by *CrGASA*s, including Cu with *cup2*∆, Zn with *zrc1*∆*cot1*∆, Ni with *smf1*∆, Mn with *pmr1*∆, and Co with *cot1*∆. None of these five *CrGASA*s could mediate Zn, Co, Ni, or Mn tolerance in yeast mutant strains (Appendix A), while only *CrGASA2* and *CrGASA14* elevated the Cu tolerance of *cup2*∆ (Figure 11D). These results indicate that the metal tolerance mediated by GASA proteins has gene-specific characteristics. Considering previous research on AtGASA5 being a metalloprotein only using iron as a metal cofactor (and not other elements, such as Ag, Al, B, Ba, and Ca) [15], we can conclude that the metal tolerance mediated by *CrGASA*s was also specific and did not have universality.

Different Arabidopsis *GASA*s have been confirmed to have the opposite effects against thermotolerance; for example, *AtGASA*4 enhanced tolerance to heat stress in transgenic overexpressing Arabidopsis [21], while *AtGASA5*-overexpressing plants displayed weaker thermotolerance than WT plants [22]. Here, we also detected yeast thermotolerance by expressing these five *CrGASA*s, and we found that *CrGASA2*, *CrGASA13*, and *CrGASA16* could obviously elevate thermotolerance both in the WT and *skn7*∆, while *CrGASA7* induced the sensitivity of yeast to heat, and *CrGASA14* only showed slightly elevated tolerance to heat stress both in WT and in the *skn7*∆ yeast strain (Figure 12). These results suggest that different *CrGASA*s might act as contrasting regulators in thermotolerance, while their concrete mechanisms require further research.

## 4. Discussion

Due to their sessile characteristics, plants are inevitably subjected to stressful challenges or pest and disease attacks throughout their life cycles. To survive these environmental disturbances and complete the alternation of generations, plants have evolved adaptive molecular mechanisms whereby the biological processes in vivo are widely protected and regulated by a variety of components, including chemical molecules, chaperones, or transcription factor regulators. The plant *GASA* family encodes a series of low-molecular-weight proteins with the characteristic of the GASA domain containing 12 conserved cysteines, which may be active redox reaction sites for key regions, regulating redox homeostasis in plants or mediating the physical interaction between GASA proteins and other proteins [25]. The objectives of this study represent the first comprehensive investigation of the *GASA* family in a specific habitat plant, *C. rosea*, and the related data undoubtedly reveal the potential roles of GASA proteins in the adaptation of plants to their extreme environmental habitats.

In model plant Arabidopsis, the *GASA* family has been proven to be involved in numerous biological processes and molecular events, including responding to surrounding or endogenous cues, regulating plant growth and development, and adapting to environmental adversities [12,16,22,27]. To date, in some crops, the *GASA* families have also been elucidated partially or systematically, mainly aiming at their critical roles in plant vegetative and reproductive organ development [20,23,25,28] or plant defense processes [39]. Although *GASA* families have been found in many plant species, the available information describes little about this gene family in wild plants from unique habitats, especially for their possible roles related to ecological adaptation in extreme environments. Here, we conducted a comprehensive genome-wide identification and expression profiling study of the *GASA* gene family in *C. rosea*. *Canavalia rosea* is widely distributed in tropical and subtropical seashores and coral islands, with high saline alkalinity, extreme drought, and continuous heat tolerance. Our research has focused on this species for several years. CrGASAs may act as protective molecules to help this plant in adapting to continuous abiotic stresses in its unique habitat.

We identified 23 *GASA* genes in the *C. rosea* genome based on the BLASTP search results (Figure 1 and Table 1). Most of the CrGASA proteins had low molecular weight and 12 conserved cysteine residues at the C-terminus, except for CrGASA12, in which the GASA domain was located in the middle of the protein (Table 1, Figure 3 and Figure 4). The CrGASAs were divided into three groups (G1, G2, and G3) based on their phylogenetic analysis with other species, including Arabidopsis, rice, and soybean (Figure 2). According to this phylogenetic tree, most CrGASAs showed closer homology with soybean GASAs, and, as a diploid dicotyledon species, *C. rosea* holds more *CrGASA* members than that in the diploid monocotyledons—for example, 14 or 10 GASA members were found in maize and rice [28,32], while almost 20 *GASA* members were identified in apple (26), citrus (18), tobacco (18), and poplars (19 or 21), which probably indicates a slight *GASA* gene expansion in dicotyledon species [35,36,37,39,40]. Because of the relatively short protein lengths and the highly conserved GASA domains of all CrGASAs (Figure 3), the classification of protein groups mostly depends on the variable N-terminus (Figure 4), which also determines the subcellular localization of CrGASAs. This issue led to the fact that, in the same group, some CrGASA members might have similar subcellular distribution patterns, such as CrGASA5, CrGASA19, CrGASA20, and CrGASA21 (G2 group, partially localized in the cell nucleus) (Figure 4, Table 1). Thus, we proposed that it might be the variability of CrGASAs’ N-terminus that accounts for their functional diversity and specificity.

According to the transcriptome data of the tissue- and habitat-specific *C. rosea* samples (Figure 6), the basic expression levels of most *CrGASA*s were not high, and they were more likely to be transcribed in tissues with better growth potential, which is consistent with the previous view that plant GASAs are primarily involved in mediating the growth and development of plants, as well as disease resistance and stress responses [27,29]. However, we presented another hypothesis that, since plant GASAs could decrease ROS accumulation to mediate hormone signaling and stress responses [15], it is reasonable that *CrGASA*s might participate in the strong adaptability of *C. rosea* plants to the extreme adversity experienced on tropical coral reefs or islands. In some previous studies, for example, *ZmGSL3* from maize and *AtGASA4* from Arabidopsis had enhanced thermotolerance when overexpressed in yeast or transgenic plants [21], while overexpression of *AtGASA5* led to heat sensitivity in plants [22]. A beechnut (*Fagus sylvatica*) GASA gene, *FsGASA4*, can improve plant resistance to salt, oxidative, and heat stress in transgenic Arabidopsis [49]. Heat stress is one of the major environmental stressors on tropical islands or reefs compared with inland areas, mainly due to the intensity of sunlight and lack of shade. Here, we only detected transcript changes in leaf samples collected in two different habitats (SCBG and YX); the expression of *CrGASA*s in the YX sample integrally presented a slightly higher level than that in the SCBG sample (Figure 6). This suggests that *CrGASA*s might participate in abiotic stress responses by regulating their expression, while the question of whether they are directly involved in thermotolerance needs to be further investigated.

Plants are subjected to various abiotic stressors, leading to the increased production of ROS that, at low levels, can act as signaling molecules in mediating stress responses in plants or, when accumulating, can cause damage to plant cells and cannot be scavenged effectively by antioxidant systems [50]. Thus, plants’ adaptation to extreme environments or climate conditions, such as heat, drought, salinity, and alkalinity, depends much more on the ROS scavenging ability, which is mainly mediated by plant antioxidant defense systems, including non-enzymatic antioxidants and antioxidant enzymes [51]. Plant GASA proteins have been confirmed to be involved in redox reactions with non-enzymatic antioxidant activities [29]; therefore, they were thought to be generally responsive to different pathogens or to mediate complex hormonal crosstalk [4], as well as to challenge various abiotic stressors during plant growth and development [19]. We also investigated the *CrGASAs*’ expression changes under different single stress challenges (salt, alkaline, high osmotic stress, and heat) by RNA-seq or qRT-PCR (Figure 7 and Figure 9), which can provide important clues for their possible involvement in *C. rosea*’s adaptability to environmental adversity. Under different stress challenges, *CrGASAs* showed spatiotemporal specificity, probably due to their involvement in different functional characterizations and perceptions of stress signaling. Several genes showed ubiquitous upregulation, especially in leaves under 48 h of challenge, such as *CrGASA4*, *CrGASA5*, *CrGASA13*, *CrGASA15*, *CrGASA16*, *CrGASA18*, and *CrGASA19*, while *CrGASA**3*, *CrGASA4*, *CrGASA7*, *CrGASA**9*, *CrGASA**12*, *CrGASA13*, *CrGASA**14*, *CrGASA16*, *CrGASA18*, *CrGASA19*, and *CrGASA23* were also partially induced in roots after 48 h of challenge. While most *CrGASAs* remained constant in roots under 2 h of salt, alkaline, or high osmotic stress, only some members (*CrGASA14* and *CrGASA16*) showed induced expression under heat stress after 2 h. In the leaf after 2 h stress challenges, most of the *CrGASAs* were obviously induced by alkaline, high osmotic and heat stresses (Figure 7 and Figure 9, and Appendix A). The complex expression patterns of *CrGASAs* under different abiotic stressors highlight their potential integral roles in various stress response processes.

Plant GASAs could also act as metalloproteins, which may be due to their conserved GASA domains holding 12 cysteine residues. Until now, there have been only a few reports about plant GASAs related to metal stress [15,28,31,48,52]. In several plant species, the expression of some *GASAs* has been induced by HMs [28,31,52]. For example, in the metal-tolerant species *Eucalyptus globulus* and *Viola baoshanensis*, *GASA* members were obviously regulated by exogenous HMs, and in rice, the *OsGASR* genes were upregulated by Cd, Cr, Ni, and Fe in gene- and spatio/temporal-specific manners. One tobacco *GASA* gene was also shown to alleviate Cd toxicity when expressed in yeast [48]. Considering that *C. rosea* is a halophyte and mangrove associate, HM tolerance and accumulation in this type of plant is also a hotspot in research about the adaptation mechanisms of plants to their native habitats. We first checked the transcriptional changes under different HM challenges (Cd, Zn, Cu, and Mn) at different time points (2 h and 48 h), and the results showed that 48 h HM stress could induce the expression of specific *CrGASA*s, such as *CrGASA16* in roots and *CrGASA4*/*CrGASA14*/*CrGASA16* in leaves, while some of the *CrGASA*s were not affected by HM challenge (Figure 8, Appendix A). Further confirmation by qRT-PCR was also performed, and the expression of *CrGASA3*, *CrGASA13*, and *CrGASA16* in the roots or *CrGASA3*, *CrGASA14*, and *CrGASA16* in the leaves was induced by metals (Figure 10). These results imply that *CrGASA*s are responsive to HMs; thus, HM stress modulates their transcription regulation network in *C. rosea* plants.

Based on these data, we propose that CrGASAs might be involved in HM detoxification, probably by their metal chelation mediated by the disulfide bonds of conserved cysteines in the GASA domain or depending on their antioxidant capacity, similar to metallothionein [53]. All five CrGASAs, unsurprisingly, showed obvious H_2_O_2_ tolerance in the yeast mutant strain *skn7*∆ (Figure 11A), indicating that these CrGASA members all presented some antioxidant activities, at least in terms of alleviating the toxicity of H_2_O_2_, which might mainly be mediated by reductive amino acids (i.e., the 12 conserved cysteines in the GASA domains). Originally, speculation about metal chelation and detoxification by these cysteines might have been consistent with different CrGASAs, but this contrasted with the yeast assays. Only CrGASA14 enhanced the Cd tolerance of WT yeast and complemented the Cd sensitivity of *ycf1*∆ (Figure 11B,C). Both CrGASA2 and CrGASA14 also complemented the Cu sensitivity of *cup2*∆ (Figure 11D). None of the five CrGASAs showed Zn, Co, Ni, or Mn tolerance in the different yeast mutant strains (Appendix A). These results are quite different from other small, cysteine-rich antioxidative proteins, such as plant metallothioneins [47,48,53], but HM detoxification mediated by metallothioneins was typical and not protein member-specific. This suggested the functional redundancy of different CrGASA members, which indicated that some CrGASA proteins could chelate active HMs, thereby alleviating HM toxicity. Moreover, this might also be due to the unique subcellular localization characteristic of specific CrGASAs, which are cysteine-rich, cytoplasmically distributed proteins, organelle- or vesicle-localized, or secretory peptides. Compared with the expression patterns of this gene family, especially the induced expression of *CrGASA2* and *CrGASA14* by HMs in RNA-seq or qRT-PCR, we can conclude that some CrGASAs undertake metal detoxification and HM phytoremediation, similar to *C. rosea* metallothioneins. However, given the highly conserved GASA domains with nearly constant cysteine residues, it is still unclear how the variable amino acid residues in GASA domains or the N-terminus affect or restrict HM chelation specificity by the disulfide bonds of conserved cysteines. Similarly, the thermotolerance or sensitivity mediated by different *CrGASA*s was also member-specific (Figure 12), indicating that the cellular heat signaling pathways mediated by or involving different *CrGASA* members were specific, or perhaps opposed, and that some *CrGASA*s do participate in the adaptability to the thermal environment in tropical regions. These findings could provide informative insights for further studies on the roles of the *CrGASA* gene family, especially in phytoremediation concerning Cd pollution or the thermotolerance of *C. rosea* plants.

## 5. Conclusions

In brief, we identified 23 *CrGASA* genes and explored their conserved motifs, tissue expression patterns, and evolutionary relationships. The expression of *CrGASA*s has the potential to be regulated by their habitats, as well as the *C. rosea* plants’ developmental stages. Therefore, *CrGASA*s were speculated to play possible roles against extreme abiotic stress. This research has also attempted to explore the biological roles of *CrGASA*s in responding to different HMs and their stress resistance. *CrGASA*s possess considerable potential for the phytoremediation of Cd and Cu. These results will help to further study plant *GASA*s and their role in abiotic stress tolerance for improving the agricultural productivity of crops.

## Figures and Tables

**Figure 1 genes-13-01988-f001:**
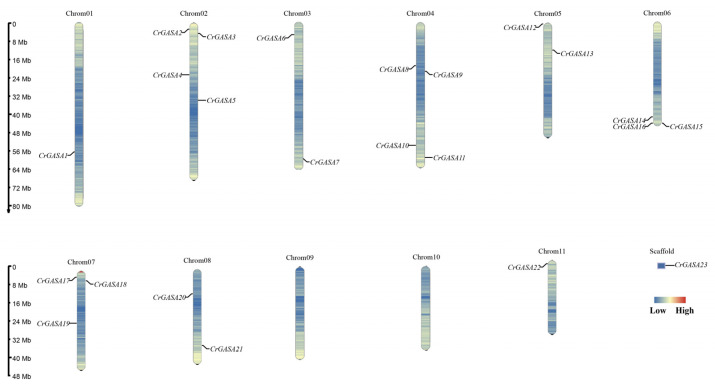
Locations of the twenty-three *CrGASA*s on eleven chromosomes and one scaffold of *C. rosea* genome.

**Figure 2 genes-13-01988-f002:**
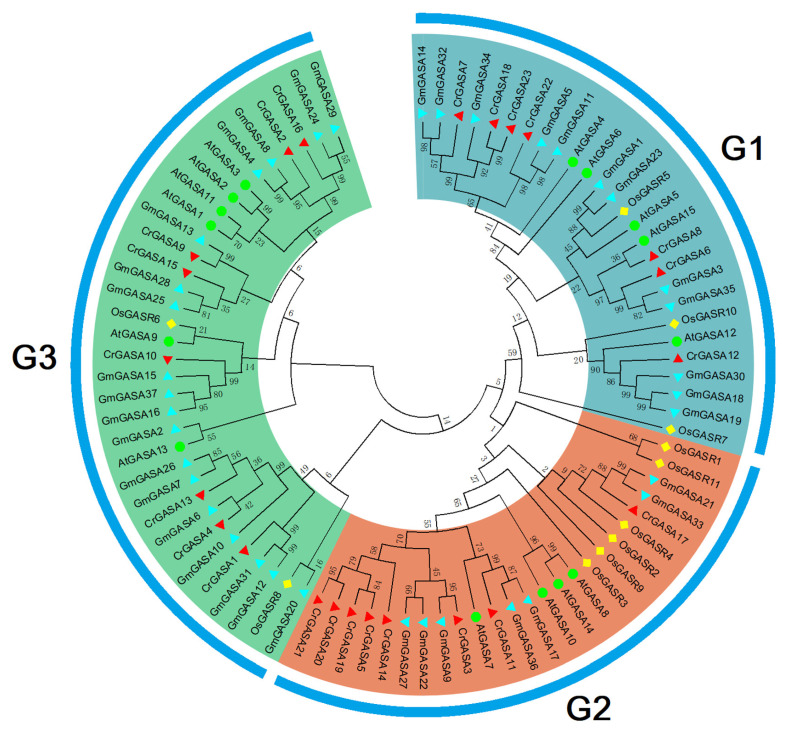
Phylogenetic relationships of the twenty-three CrGASAs from *C. rosea*, fifteen AtGASAs from *Arabidopsis thaliana*, eleven OsGASRs from *Oryza sativa*, and thirty-seven GmGASAs from *Glycine max*. These eighty-six amino acid sequences of four plant species were compared with ClustalW alignment, and the phylogenetic tree was constructed in MEGA 6.0 using the neighbor-joining method, with 1000 bootstrap repetitions. The different branch colors represent different groups, including G1 (cyan), G2 (brown), and G3 (green).

**Figure 3 genes-13-01988-f003:**
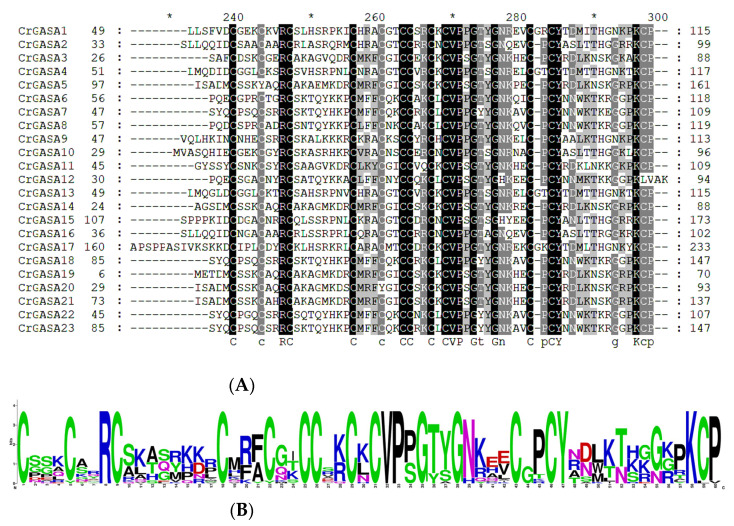
Alignment of the GASA domain from CrGASA proteins. (**A**) Multiple alignments of the CrGASA protein sequences. Their conserved GASA domains are indicated. (**B**) Sequence logo analysis of the conserved GASA domains. Each stack represents their amino acids.

**Figure 4 genes-13-01988-f004:**
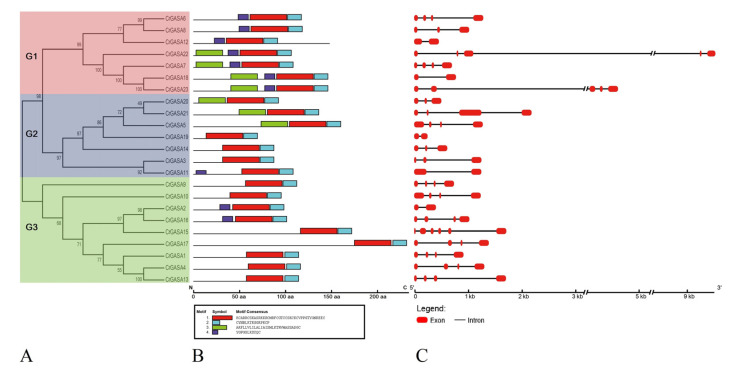
Structural analysis of the CrGASA proteins and genes. (**A**) The phylogenetic tree constructed using MEGA 6.0. The three major groups are marked with different background colors. (**B**) The conserved motifs of each group identified by the MEME web server. Different motifs are represented by different colored boxes, and the motif sequences are provided at the bottom. (**C**) The exon–intron organization of the CrGASAs constructed using GSDS 2.0.

**Figure 5 genes-13-01988-f005:**
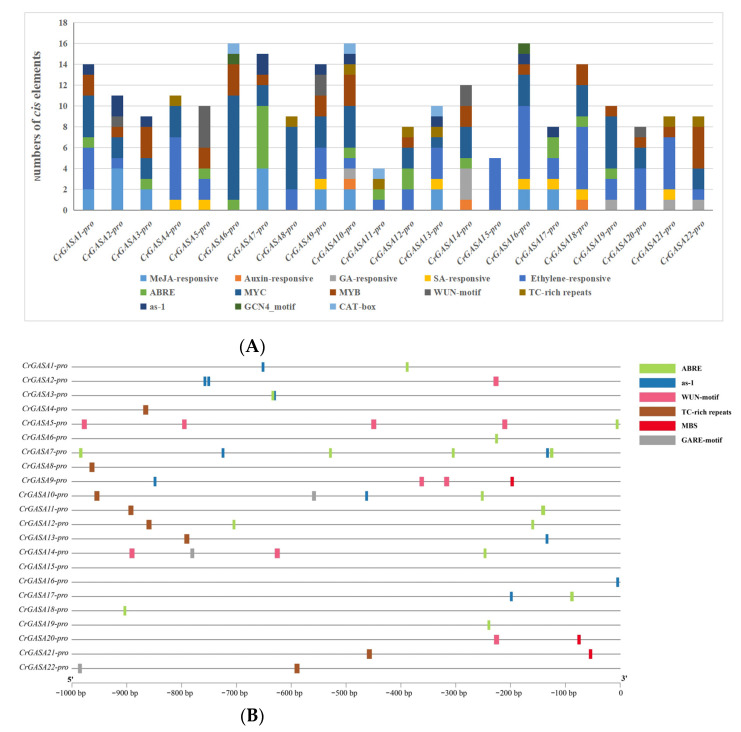
Statistics for predicted *cis*-regulatory elements in the *CrGASA* promoters (ATG upstream 1000). (**A**) Summaries of the thirteen *cis*-regulatory elements in the twenty-two *CrGASA* promoter regions. (**B**) Distribution of the six *cis*-regulatory elements (ABRE, as-1, WUN motif, TC-rich repeats, MBS, gibberellin-responsive motif, GARE) in the twenty-two *CrGASA* promoter regions. The elements are represented by different symbols. The scale bar represents 100 bp.

**Figure 6 genes-13-01988-f006:**
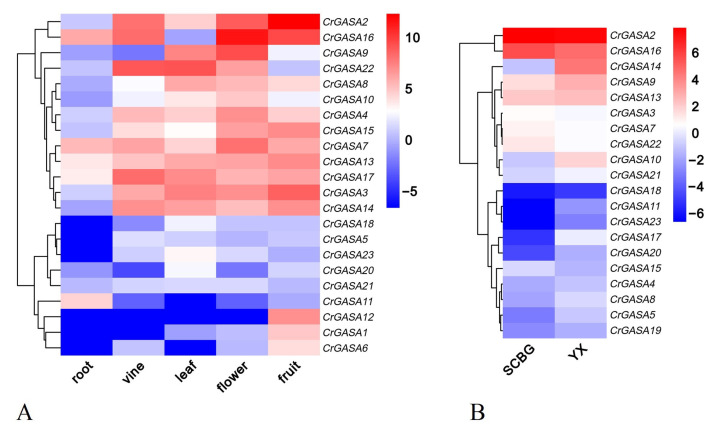
Heatmaps showing (**A**) the expression levels of the *CrGASA*s in the root, vine, leaf, flower bud, and young fruit of *C. rosea* plants and (**B**) expression differences of the *CrGASA*s in mature *C. rosea* leaves planted in the South China Botanical Garden (SCBG) and in Yongxing (YX) Island. The expression level of each gene is shown in FPKM (log2). Red denotes high expression levels, and blue denotes low expression levels. Data normalization was performed on the Oebiotech Cloud website (https://cloud.oebiotech.cn/task/detail/heatmap/ (accessed on 1 March 2022)).

**Figure 7 genes-13-01988-f007:**
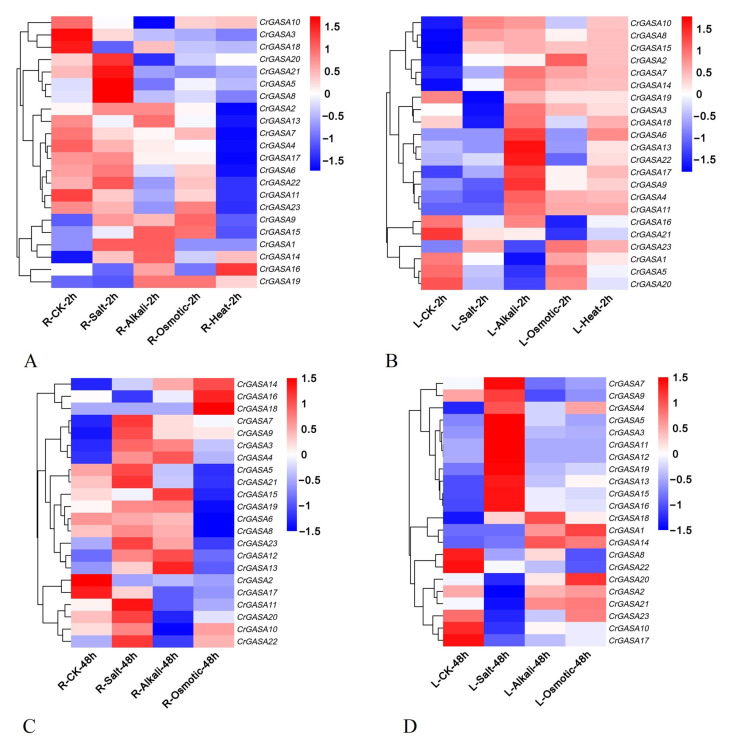
Heatmaps showing the expression changes of *CrGASA*s under high salinity, alkaline, high osmotic stresses, and heat challenge. (**A**,**B**) The expression differences of *CrGASA*s in the root (**A**) and leaf (**B**) after 2 h abiotic stress challenges; (**C**,**D**) the expression differences of *CrGASA*s in the root (**C**) and leaf (**D**) after 48 h abiotic stress challenges. CK: control. The expression level of each gene is shown in FPKM (log2). Red denotes high expression levels, and blue denotes low expression levels. Data normalization was performed on the Oebiotech Cloud website (https://cloud.oebiotech.cn/task/detail/heatmap/ (accessed on 1 March 2022)).

**Figure 8 genes-13-01988-f008:**
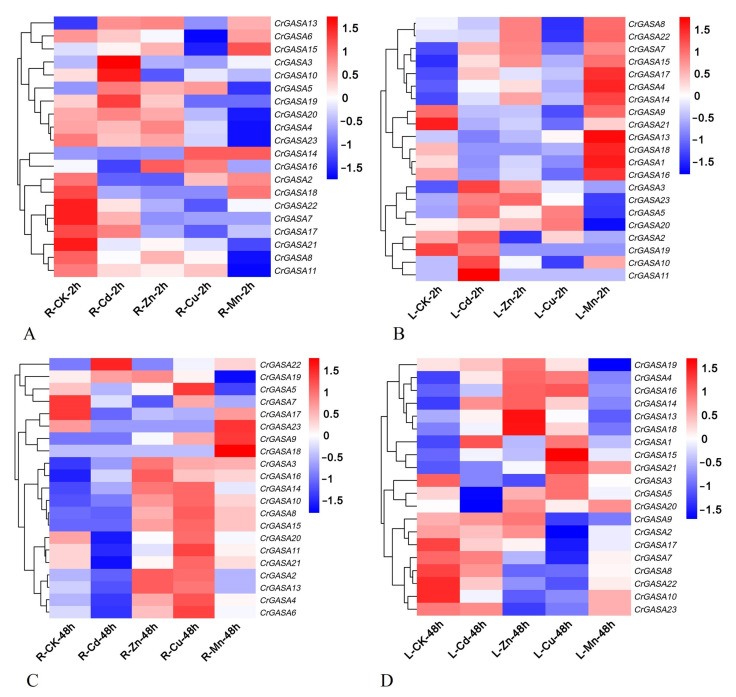
Heatmaps showing the expression levels of the *CrGASA*s under heavy metal stress, including cadmium (Cd), zinc (Zn), copper (Cu), and manganese (Mn). (**A**,**B**) The expression differences of *CrGASA*s in the root (A) and leaf (B) after 2 h heavy metal stress; (**C**,**D**) the expression differences of *CrGASA*s in the root (**C**) and leaf (**D**) after 48 h heavy metal stress. CK: control. The expression level of each gene is shown in FPKM (log2). Red denotes high expression levels, and blue denotes low expression levels. Data normalization was performed on the Oebiotech Cloud website (https://cloud.oebiotech.cn/task/detail/heatmap/ (accessed on 1 March 2022)).

**Figure 9 genes-13-01988-f009:**
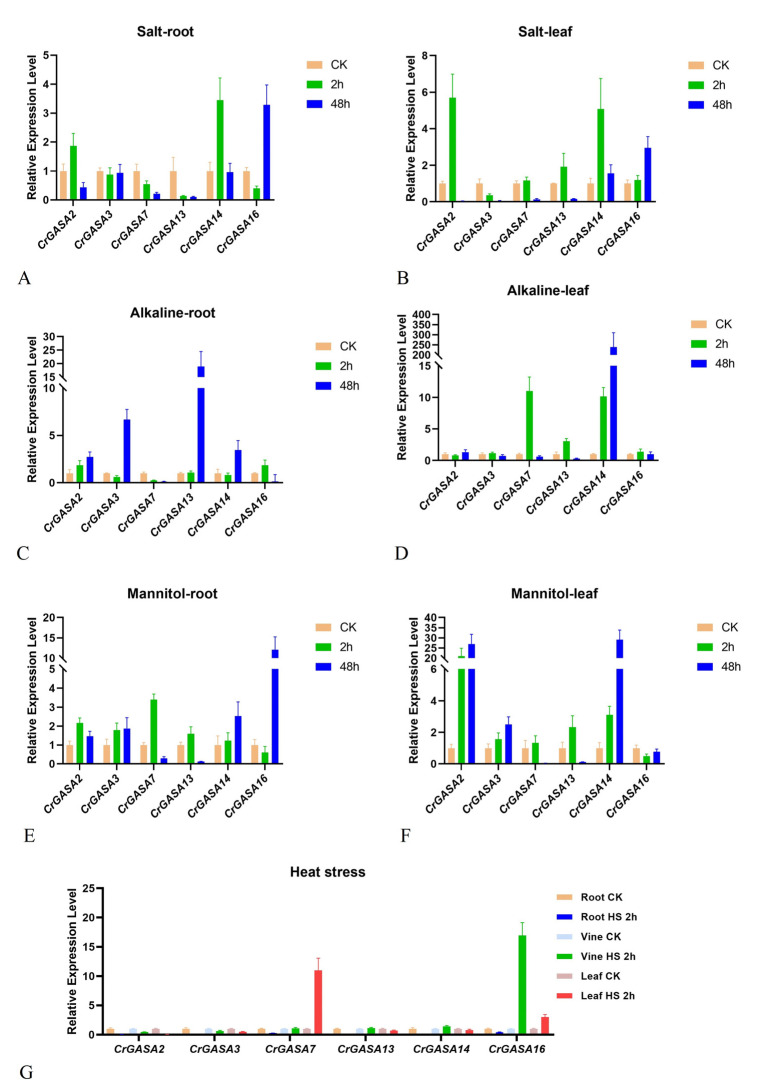
Quantitative RT-PCR detection of the expression levels of the six *CrGASA*s in *C. rosea* seedlings responding to different stresses, including 600 mM NaCl, 150 mM NaHCO_3_, 300 mM mannitol, and heat stress (45 °C), for different times (0, 2 h, and 48 h). (**A**,**B**) Root and leaf samples under 600 mM NaCl treatment; (**C**,**D**) root and leaf samples under 150 mM NaHCO_3_ treatment; (**E**,**F**) root and leaf samples under 300 mM mannitol treatment; and (**G**) root, vine, and leaf samples under 2 h heat stress challenge. CK: control. Relative expression values were calculated using the 2^−ΔCt^ method, with the housekeeping gene *CrEF-1α* as a reference gene. Bars show the mean values ± SD of *n* = 3–4 technical replicates.

**Figure 10 genes-13-01988-f010:**
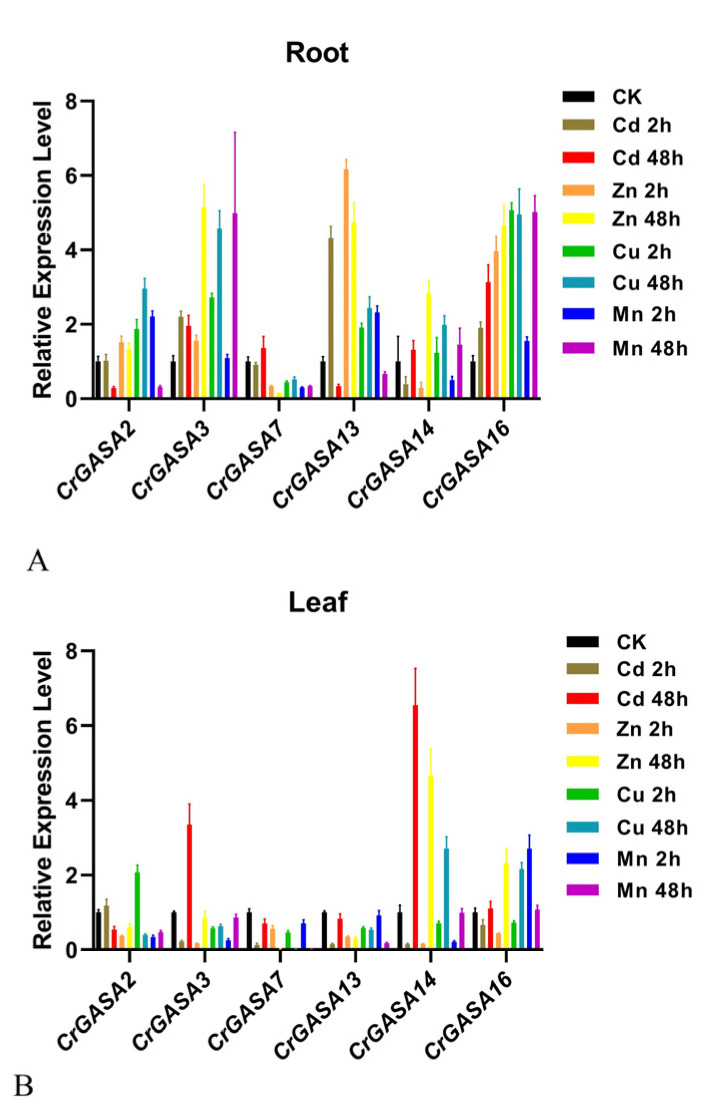
Quantitative RT-PCR detection of the expression levels of the six *CrGASA*s responding to different heavy metal (HM) stresses (0.1 mM CdCl_2_, 0.5 mM ZnSO_4_, 1 mM MnCl_2_, and 0.1 mM CuSO_4_) in *C. rosea* seedlings. (**A**) Root samples under different HM stresses; (**B**) leaf samples under different HM stresses. CK: control. Relative expression values were calculated using the 2^−ΔCt^ method, with the housekeeping gene *CrEF-1α* as a reference gene. Bars show the mean values ± SD of *n* = 3–4 technical replicates.

**Figure 11 genes-13-01988-f011:**
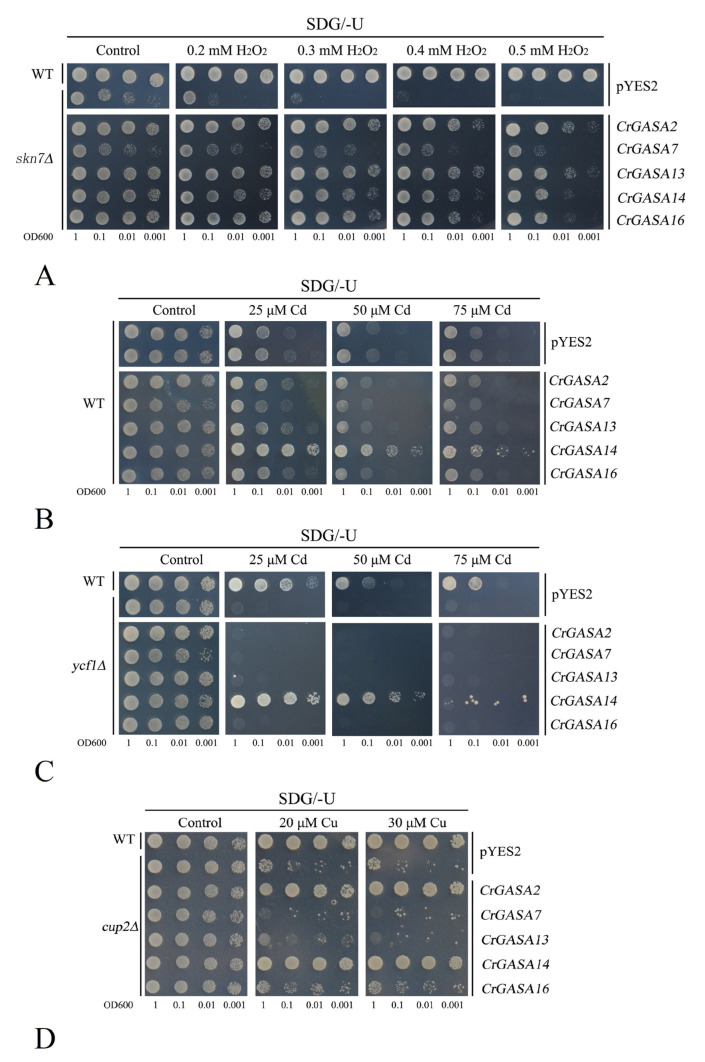
Functional identification of five *CrGASA*s in yeast by heterologous expression assay. The yeast wild-type (WT) strain BY4741, and yeast mutant strains *skn7*Δ (sensitive to H_2_O_2_), *ycf1*Δ (sensitive to cadmium), and *cup2*Δ (sensitive to copper), were transformed with the empty vector pYES2 and five recombinant vectors, including CrGASA2-pYES2, CrGASA7-pYES2, CrGASA13-pYES2, CrGASA14-pYES2, and CrGASA16-pYES2. Yeast cultures were adjusted to OD600 = 1, and 2 μL serial dilutions (10-fold, from left to right in each panel) were spotted on SDG-Ura medium supplemented with different concentrations of stressors. (**A**) H_2_O_2_ oxidative stress tolerance confirmation in yeast mutant strain *skn7*Δ; (**B**) cadmium tolerance confirmation in WT yeast; (**C**) cadmium tolerance confirmation in mutant yeast strain *ycf1*Δ; (**D**) copper tolerance confirmation in mutant yeast strain *cup2*Δ.

**Figure 12 genes-13-01988-f012:**
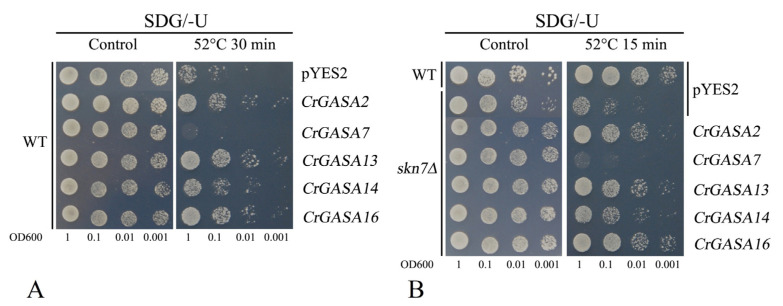
The thermotolerance confirmation in yeast strains WT (**A**) and *skn7*Δ (**B**) by expressing five *CrGASA*s. Yeast cultures (with or without heat stress challenges, WT: 52 °C for 30 min; *skn7*Δ: 52 °C for 15 min) were adjusted to OD600 = 1, and 2 μL serial dilutions (10-fold) were spotted on SDG-Ura medium plates. The plates were incubated for 2–5 days at 30 °C.

**Table 1 genes-13-01988-t001:** Nomenclature and subcellular localization of CrGASAs identified from *C. rosea* genome.

Name	Locus	Protein Length	Major Amino Acid%	Mw (kD)	PI	II	AI	GRAVY	TMHs and Topologies	Subcellular Localization
CrGASA1	01T002406	115	C (11.3%), L (10.4%), K (9.6%)	12.83	9.32	37.75	71.13	–0.33	1/in to out	extr: 8, vacu: 4, cyto: 1
CrGASA2	02T004079	99	C (12.1%), L (11.1%), A (10.1%)	10.73	8.84	51.61	79.9	0.040	1/in to out	extr: 11, vacu: 2
CrGASA3	02T004275	88	C (14.8%), K (12.5%), G (8.0%)	9.61	8.82	31.5	57.61	0.032	None	extr: 12, golg: 2
CrGASA4	02T005653	117	L (12.0%), C (11.1%), R8.5 (%)	12.95	8.24	47.68	84.02	–0.079	1/in to out	extr: 9, vacu: 3, nucl: 1
CrGASA5	02T006081	161	S (13.0%), L (9.3%), K (9.3%)	17.51	9.44	29.22	73.42	–0.208	None	nucl: 9, chlo: 3, cyto: 1
CrGASA6	03T007841	118	C (11.0%), L (10.2%), K (9.3%)	13.12	9.12	51.68	69.41	–0.258	1/in to out	extr: 5, nucl: 4, vacu: 2, cyto: 1, mito: 1
CrGASA7	03T010634	109	C (11.0%), K (11.0%), G (7.3%)	12.23	9.29	41.85	54.59	–0.236	1/in to out	extr: 10, chlo: 3
CrGASA8	04T011946	119	P (11.8%), C (10.9%), A (10.1%)	12.67	9.12	42.58	60.76	–0.202	1/in to out	extr: 10, chlo: 2, cyto: 1
CrGASA9	04T011999	113	K (11.5%), C (10.6%), L (8.8%)	12.68	9.31	52.40	71.68	–0.307	1/in to out	extr: 13
CrGASA10	04T013414	96	C (13.5%), L (6.2%), K (6.2%)	10.53	8.85	53.80	70.10	–0.109	1/in to out	extr: 10, golg: 2, cyto: 1
CrGASA11	04T013838	109	C (13.8%), L (11.0%), S (11.0%)	12.08	9.06	26.61	68.90	–0.047	1/in to out	chlo: 12, extr: 2
CrGASA12	05T014257	149	C (9.4%), A (9.4%), K (8.7%)	16.55	9.56	37.33	68.05	–0.204	None	extr: 12, vacu: 2
CrGASA13	05T015099	115	C (11.3%), L (11.3%), G (10.5%)	12.62	8.44	42.52	77.04	–0.183	1/in to out	extr: 12, vacu: 2
CrGASA14	06T019101	88	C (13.6%), S (12.5%), K (11.4%)	9.67	9.40	30.93	56.59	–0.045	1/in to out	extr: 7, chlo: 4, vacu: 2.5, E.R._vacu: 2
CrGASA15	06T019306	173	C (9.2%), S (9.2%), L (8.7%)	18.88	8.71	62.17	90.69	0.175	1/in to out	plas: 5, golg: 4, chlo: 2, E.R.: 2
CrGASA16	06T019307	102	L (14.7%), C (11.8%), A (8.8%)	10.89	9.08	45.13	88.04	–0.056	1/in to out	extr: 7, chlo: 4, E.R._plas: 2, plas: 1.5
CrGASA17	07T019638	233	P (25.8%), V (10.7%), K (9.9%)	24.55	9.71	80.18	70.21	–0.225	None	extr: 12, vacu: 1
CrGASA18	07T019724	147	L (11.6%), C (8.8%), G (8.2%)	16.44	9.34	57.44	83.47	–0.270	2/out to out	plas: 5, E.R.: 3, extr: 2, vacu: 2, cyto: 1
CrGASA19	07T020256	70	C (18.6%), K (12.9%), S (8.6%)	7.86	9.15	25.60	25.14	–0.591	None	nucl: 9, mito: 2, extr: 2
CrGASA20	08T021709	93	K (12.9%), C (11.8%), S (10.8%)	10.21	9.41	38.59	44.19	–0.555	None	nucl: 12, chlo: 1
CrGASA21	08T022646	137	C (10.2%), K (10.2%), S (10.2%)	15.24	9.38	34.29	52.77	–0.244	1/in to out	nucl: 10, chlo: 3
CrGASA22	11T027756	107	C (11.2%), K (9.3%), L (8.4%)	11.75	9.23	34.55	64.77	–0.147	1/in to out	extr: 13
CrGASA23	UnT032831	147	L (11.6%), C (8.8%), G (8.2%)	16.41	9.34	57.44	82.18	–0.254	2/out to out	plas: 4, vacu: 3, E.R.: 3, extr: 2, cyto: 1

MW: molecular weight; PI: isoelectric point; II: instability index; AI: aliphatic index; GRAVY: grand average of hydropathicity. The molecular weight and isoelectric points of predicted CrGASAs were detected using the ExPASy proteomics server (https://web.expasy.org/protparam/ (accessed on 1 March 2022)). The TMHMM Server 2.0 program (http://www.cbs.dtu.dk/services/TMHMM/ (accessed on 1 March 2022)) and the Protein Fold Recognition Server tool (PHYRE^2^, http://www.sbg.bio.ic.ac.uk/phyre2/html/page.cgi?id= index (accessed on 1 March 2022)) were used to predict the transmembrane helices and for the 3D prediction of CrGASAs, i.e., the “TMHs and Topologies” column. For the subcellular localization prediction, the online program WoLF_PSORT (https://www.genscript.com/wolf-psort.html (accessed on 1 March 2022)) was used.

**Table 2 genes-13-01988-t002:** Ka/Ks analysis and duplicated type calculation for *CrGASA* genes.

Duplicated Pair	Duplicate Type	Ka	Ks	Ka/Ks	*p*-Value (Fisher)	Positive Selection
*CrGASA2-CrGASA16*	Segmental	0.135321	0.587267	0.230425	3.22 × 10^−7^	No
*CrGASA3-CrGASA14*	Segmental	0.169573	0.654909	0.258926	1.04 × 10^−5^	No
*CrGASA4-CrGASA13*	Segmental	0.0755763	0.929139	0.0813401	3.44 × 10^−18^	No
*CrGASA6-CrGASA8*	Segmental	0.180004	0.748689	0.240425	3.69 × 10^−8^	No
*CrGASA7-CrGASA18*	Segmental	0.0790249	0.780183	0.10129	5.13 × 10^−14^	No
*CrGASA15-CrGASA16*	Tandem	\	\	\	\	\

## Data Availability

Not applicable.

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
