# Peer review of "Genome-Wide Identification and Functional Analysis of the GASA Gene Family Responding to Multiple Stressors in Canavalia rosea"

_genes, 2022, doi:10.3390/genes13111988_

Round 1

Reviewer 1 Report

Manuscript entitled "Genome-wide identification and functional analysis of the GASA gene family responding to multiple stressors in Canavalia rosea" provides comprehensive information about GASA gene family and their functionality and importance for physiological processes. Obviously, the authors made a lot of work and the overall quality of the manuscript is high. Therefore, I have only several suggestions/questions for the improvement of the manuscript.

[line 121] I think the sentence starting with "Canavalia rosea is a perennial..." should start new paragraph. It makes more sense in my opinion.

[line 227] Use shortcut "qRT-PCR" instead of "QRT-PCR" even it is at the beginning of the sentence because it is the name of the method (same "problem" in [line 470]). Further, this methodological section is very brief (no information about used amplification conditions, instrument etc.). Next, in the Results section is mentioned that delta delta Ct method was used [line 491]. It is very common approach, but it works only if the efficiency of used primers is similar. Did you check it?

[line 251 and related lines in the Results] What kind of statistical analyses do you mean? Explicitly mentioned was only Fisher´s exact test and other statistical results are not mentioned neither in the Results nor in the Discussion part. Moreover, many results are merely descriptive without any formal comparison via some statistical test (for example relative expression of selected genes under various conditions). This paragragh is uninformative and potentially important information is missing.

[line 292] Figure 1. Resolution of the picture is low and text is difficult to read. Maybe it is caused by compression into pdf format, but I should mention it. Moreover, what is the meaning of accompanied legend with "Low-High" bar? There is no explanation.

[line 295-307] This phylogenetic analysis deserves more attention. In the Material and Methods section is mentioned that NJ method with 1000 bootstraps was used. I´m not expert in the phylogenetic field, but as far as I know it is common that reliable support for the bootstrap is above 70% and I suppose for the phylogenetic studies is the treshold probably even higher (above 90%?). I can see on the Figure 2 very low bootstrap support for the basal branches. Therefore, tree topology and the informative value of this result is questionable. Despite this potential pitfall the authors did not mention it or discuss it. Further, what was the underlying principle behind the definition of these 3 selected groups? Was it defined manually or by program set-up? When I checked the Figure 2 there could be 3 little bit different groups or even four groups. Have definition of these 3 groups some hidden reason? It took my interest because when I checked the other "GASA gene articles" all articles with no exception defined exactly 3 groups of GASA genes.

[line 350] Figure 4. Same problem with the resolution quality as in Figure 1. Text is almost unreadable.

Figure 7 [line 442], Figure 8 [line 462], Figure 9 [486] and Figure 10 [line 521] - abbreviation for CK is missing. I suppose (and hope) CK is probably some abbreviation for "control" group.

[line 671]  In the sentence "...was induced by metals (Fig. 9)" authors probably reffer to Fig. 10 instead of Fig. 9.

[-] I was not abble to check Supplementary Materials.

Author Response

Manuscript entitled "Genome-wide identification and functional analysis of the GASA gene family responding to multiple stressors in Canavalia rosea" provides comprehensive information about GASA gene family and their functionality and importance for physiological processes. Obviously, the authors made a lot of work and the overall quality of the manuscript is high. Therefore, I have only several suggestions/questions for the improvement of the manuscript.

Response: Great thanks for your recognition. We have incorporated changes that reflect the detailed suggestions you have graciously provided, and we hope to meet with your approval. Revised portion are marked in red in the manuscript. The main corrections and the responds to your comments are listed in the follows.

[line 121] I think the sentence starting with "Canavalia rosea is a perennial..." should start new paragraph. It makes more sense in my opinion.

Response: Thanks. We have revised it, and set a new paragraph here.

[line 227] Use shortcut "qRT-PCR" instead of "QRT-PCR" even it is at the beginning of the sentence because it is the name of the method (same "problem" in [line 470]). Further, this methodological section is very brief (no information about used amplification conditions, instrument etc.). Next, in the Results section is mentioned that delta delta Ct method was used [line 491]. It is very common approach, but it works only if the efficiency of used primers is similar. Did you check it?

Response: Thanks for your reminder. In the "Materials and Methods" part for "Expression pattern analysis using qRT-PCR", we have added some expression about the qRT-PCR methods, and some related agents and instrument.

[line 251 and related lines in the Results] What kind of statistical analyses do you mean? Explicitly mentioned was only Fisher´s exact test and other statistical results are not mentioned neither in the Results nor in the Discussion part. Moreover, many results are merely descriptive without any formal comparison via some statistical test (for example relative expression of selected genes under various conditions). This paragragh is uninformative and potentially important information is missing.

Response: We added the expression in result part as: “Six CrGASAs were selected for qRT-PCR verification assays, and these assays were all repeated for three biological replicates. Here the figures showed only one typical experimental result of these biological replicates, which were all roughly consistent with the RNA-seq data”.

[line 292] Figure 1. Resolution of the picture is low and text is difficult to read. Maybe it is caused by compression into pdf format, but I should mention it. Moreover, what is the meaning of accompanied legend with "Low-High" bar? There is no explanation.

Response: Sorry for the poor-quality images of this manuscript. We have replaced them with fine images. They were automatically compressed when be inserted into the manuscript, and we adjusted this set.

[line 295-307] This phylogenetic analysis deserves more attention. In the Material and Methods section is mentioned that NJ method with 1000 bootstraps was used. I´m not expert in the phylogenetic field, but as far as I know it is common that reliable support for the bootstrap is above 70% and I suppose for the phylogenetic studies is the threshold probably even higher (above 90%?). I can see on the Figure 2 very low bootstrap support for the basal branches. Therefore, tree topology and the informative value of this result is questionable. Despite this potential pitfall the authors did not mention it or discuss it. Further, what was the underlying principle behind the definition of these 3 selected groups? Was it defined manually or by program set-up? When I checked the Figure 2 there could be 3 little bit different groups or even four groups. Have definition of these 3 groups some hidden reason? It took my interest because when I checked the other "GASA gene articles" all articles with no exception defined exactly 3 groups of GASA genes.

Response: The plant GASA families were firstly classified by (R Zimmermann et al., Plant Physiol, 2010, DOI: 10.1104/pp.109.149054), with 14 Arabidopsis AtGASAs, 9 rice OsGASRs (or OsGSRs/OsGSLs), 10 maize ZmGSLs, and 2 tomato SlGASTs. They were clearly classified into three subfamilies. Then in apple (BMC Genomics, 2017, 18: 827, DOI: 10.1186/s12864-017-4213-5), soybean (Plant Mol Biol, 2019, 100: 607–620, DOI: 10.1007/s11103-019-00883-1), wheat (Front Genet, 2019, 10: 980, DOI: 10.3389/fgene.2019.00980), grapevine (Int J Mol Sci, 2020, 21, 1088, DOI: 10.3390/ijms21031088), Citrus clementina (BMC Plant Biol, 2021, 21: 565, DOI: 10.1186/s12870-021-03326-6), two poplars (Int J Mol Sci, 2021, 22, 12336, DOI: 10.3390/ijms222212336; Int J Mol Sci, 2022, 23, 1507, DOI: 10.3390/ijms23031507), and tobacco (Front Genet, 2022, 12: 768942, DOI: 10.3389/fgene.2021.768942), they were all classified into three subfamilies or groups. With one exception, the rice OsGASRs were classified into four subgroups (J Plant Physiol, 2019, 234–235: 117–132, DOI: 10.1016/j.jplph.2019.02.005). Here we mainly referenced the soybean’s classification (Plant Mol Biol, 2019, 100: 607–620). And also, we found that if we only listed the Canavalia rosea CrGASAs for the phylogenetic analysis, the classification information would be very clear, and there were grouped into three branches as the following:

[line 350] Figure 4. Same problem with the resolution quality as in Figure 1. Text is almost unreadable.

Response: We re-inserted all of figures with the high-quality images. Sorry about that.

Figure 7 [line 442], Figure 8 [line 462], Figure 9 [486] and Figure 10 [line 521] - abbreviation for CK is missing. I suppose (and hope) CK is probably some abbreviation for "control" group.

Response: We added the “CK: control.” in the figure legend parts of figures 7, 8, 9, and 10.

[line 671]  In the sentence "...was induced by metals (Fig. 9)" authors probably refer to Fig. 10 instead of Fig. 9.

Response: Thanks for your reminder. We revised it.

[-] I was not able to check Supplementary Materials.

Response: The “Supplementary Materials” have been uploaded entirely.

Reviewer 2 Report

Line number 25-26 need more clarification 

Author Response

Line number 25-26 need more clarification

Response: Here the “overexpression” means the induced expression of CrGASAs in yeast cell by the 2% galactose (the sole carbon source) of the SDG medium, and we revised this description as: “Heterologous induced expression...”. The related formulation was also added in the “Materials and Methods” and “Results” parts.
